# Herbicide-Resistance in Turf Systems: Insights and Options for Managing Complexity

**Jennifer H. Allen** [1,*], **David E. Ervin** [2], **George B. Frisvold** [3], **James T. Brosnan** [4], **James D. McCurdy** [5], **Rebecca G. Bowling** [6], **Aaron J. Patton** [7], **Matthew T. Elmore** [8], **Travis W. Gannon** [9], **Lambert B. McCarty** [10], **Patrick E. McCullough** [11], **John E. Kaminski** [12], **Shawn D. Askew** [13], **Alec R. Kowalewski** [14], **J. Bryan Unruh** [15], **J. Scott McElroy** [16] **and Muthukumar V. Bagavathiannan** [17]

1   Department of Public Administration, Portland State University, Portland, OR 97201, USA
2   Department of Environmental Science and Management, Institute for Sustainable Solutions, Department of Economics, Portland State University, Portland, OR 97201, USA
3   Department of Agricultural & Resource Economics, University of Arizona, Tucson, AZ 85721, USA
4   Department of Plant Sciences, University of Tennessee, Knoxville, TN 37996, USA
5   Department of Plant and Soil Sciences, Mississippi State University, Starkville, MS 39762, USA
6   Department of Soil & Crop Sciences, Texas A&M AgriLife Center-Dallas, Dallas, TX 75252, USA
7   Department of Horticulture & Landscape Architecture, Purdue University, West Lafayette, IN 47907, USA
8   Department of Plant Biology, Rutgers University, New Brunswick, NJ 08901, USA
9   Department of Crop and Soil Sciences, North Carolina State University, Raleigh, NC 27695, USA
10  Department of Plant, & Environmental Sciences, Clemson University, Clemson, SC 29634, USA
11  Department of Crop & Soil Sciences, University of Georgia, Griffin, GA 30223, USA
12  Department of Plant Science, Pennsylvania State University, University Park, PA 16802, USA
13  School of Plant & Environmental Sciences, Virginia Polytechnic Institute & State University, Blacksburg, VA 24061, USA
14  Department of Horticulture, Oregon State University, Corvallis, OR 97331, USA
15  Environmental Horticulture Department, West Florida Research and Education Center, Institute of Food and Agricultural Sciences, University of Florida, Jay, FL 32565, USA
16  Department of Crop, Soil, & Environmental Sciences, Auburn University, Auburn, AL 36849, USA
17  Department of Soil and Crop Sciences, Texas A&M University, College Station, TX 77843, USA
*   Correspondence: jhallen@pdx.edu

**Abstract:** Due to complex interactions between social and ecological systems, herbicide resistance has classic features of a "wicked problem." Herbicide-resistant (HR) *Poa annua* poses a risk to sustainably managing U.S. turfgrass systems, but there is scant knowledge to guide its management. Six focus groups were conducted throughout the United States to gain understanding of socio-economic barriers to adopting herbicide-resistance management practices. Professionals from major turfgrass sectors (golf courses, sports fields, lawn care, and seed/sod production) were recruited as focus-group participants. Discussions emphasized challenges of the weed management of turfgrass systems as compared to agronomic crops. This included greater time constraints for managing weeds and more limited chemical control options. Lack of understanding about the proper use of compounds with different modes of action was identified as a threat to sustainable weed management. There were significant regional differences in perceptions of the existence, geographic scope, and social and ecological causes of HR in managing *Poa annua*. Effective resistance management will require tailoring chemical and non-chemical practices to the specific conditions of different turfgrass sectors and regions. Some participants thought it would be helpful to have multi-year resistance management programs that are both sector- and species-specific.

**Keywords:** turfgrass managers; integrated weed management; wicked problems; focus groups; socio-economic surveys; annual bluegrass; *Poa annua* L.

## 1. Introduction

Developing management strategies to address challenges that are complex, evolving, multi-causal and laden with uncertainty– often termed "wicked problems"—requires a thoughtful exploration of the multi-faceted nature of these issues [1]. Some core attributes of wicked problems, from [2], are as follows:

- Difficult to clearly define;
- Have many interdependencies and may be multi-causal;
- Are unstable and continuously evolving;
- Have no clear solution;
- Involve multiple stakeholders with differing ideas about what the "real" problem is as well as what the causes of problem may be;
- Do not fit within the responsibility of one organization;
- May require changing behavior to resolve;
- Characterized by chronic policy failure.

Natural-resource management challenges that involve "common pool" resources share many of the characteristics of wicked problems, in particular the level of interdependencies, involvement of multiple stakeholders, not fitting into the responsibility of any one organization, and the need for behavior change to address them [3,4]. The literature on both wicked problems and common-pool resources offers insights that can help in better understanding the dynamics of these issues as well as informing strategies that can be used to address them.

Herbicide resistance has been characterized as "the epitome of a wicked problem: the causes are convoluted by myriad biological and technological factors, and are fundamentally driven by the vagaries of human decision-making" [5] (p. 552), [6–9]). Common-pool problems in herbicide-resistance management are also increasingly acknowledged by social scientists [6,10–12]. Ervin and Frisvold [10] note it is in the collective interest of farmers to delay resistance and conserve the usefulness of an herbicide and herbicide mechanism of action, but it can be costly for individual farms to adopt alternative weed-control tactics to delay resistance. Delaying resistance, they argue, thus "becomes a 'common-pool' problem—each farmer has an individual incentive to use the herbicide in the short run without considering effects on resistance." [10] (p. 610).

Jussaume and Ervin [7] note that recognizing herbicide resistance as a wicked problem is "necessary for scientists and practitioners, who will need to use adaptive management principles collaboratively to explore the efficacy of an integrated set of alternative management approaches for managing weeds" [7] (p. 2). These authors further assert that developing strategies to successfully address weed resistance requires "a shared perspective that incorporates an improved understanding of the human dimensions of weed management" [7] (p. 2). As Ervin and Frisvold [10] note, the "common-pool nature of weed susceptibility means that managing resistance is not just an agronomic problem, but a social one as well. A fundamental question then becomes, "what type of social organizations can effectively manage common-pool resources?" [10] (p. 610).

The research presented here seeks to tap the experiential knowledge of turf managers and their advisers to start building an understanding of the socio-ecological factors influencing the management of herbicide resistance in annual bluegrass (*Poa annua* L.). Annual bluegrass (*Poa annua* L.) was ranked as the third most common as well as the most troublesome turfgrass weed in the United States and Canada in a recent survey [13] and has been found on all continents globally, including Antarctica [14]. The widespread distribution of annual bluegrass is related to the species' ability to tolerate a wide range of environmental conditions. This research is part of a larger multi-state project supported by the USDA-NIFA Specialty Crops Research Initiative (SCRI) undertaken by a team of 16 universities to limit the impacts of annual *Poa annua* in athletic, golf, lawn, and sod-farm turfgrass (http://resistpoa.org/ (accessed on 1 October 2022)). The research presented here addresses two specific objectives of this larger project: to identify socio-economic constraints affecting control strategies and stewardship of herbicide technologies, and to

develop and deliver best management practices (BMPs) through cooperative extension and education.

The central purpose of this research is to better understand the dynamics related to herbicide resistance in *Poa annua* and to explore the motivations and challenges in fostering more integrated weed management in turfgrass systems to control the threat posed by increasing herbicide resistance. This research offers several novel contributions. In general, herbicide resistance management in turfgrass systems is less well understood than resistance in agronomic crops; furthermore, while focus-group methods have been usefully employed to examine resistance management for agronomic crops, this has not been performed for turfgrass systems. This research addresses these gaps and contributes to a broader and deeper understanding of herbicide resistance in turfgrass systems.

Understanding the "wicked" elements of this challenge may assist in the identification of strategies to address the development of herbicide resistance in *Poa annua*. While Brosnan et al. [15] have recently begun to explore herbicide-resistance management in turfgrass systems as a "wicked problem", this publication adds to the sparse literature on social decisions associated with turfgrass weed management and herbicide resistance and may help identify approaches to address the complex challenge of managing herbicide-resistant *Poa annua*.

## 2. Background and Methods

Turfgrass systems provide a broad range of environmental, economic and social ecosystem services [16–19]. Some examples include temperature moderation, soil stabilization which can retard surface runoff and enhance stormwater nutrient retention and water infiltration, and supporting recreation activities and related physical and mental health benefits [20,21]. Reviews of the value of specific services such as reduced heating and cooling costs from temperature moderation, property value enhancements from aesthetically pleasing landscapes, reduced maintenance costs, and sports/recreation benefits indicate that turfgrass systems make significant contributions to economic welfare [19,22–29].

The occurrence of herbicide-resistant weeds, such as *Poa annua*, can degrade turfgrass quality and diminish various benefits over time. For example, weeds have the potential to increase risks of athletic injuries on playing fields [30]. The presence of weeds can reduce golf-course aesthetics, leading to lower golf-course revenues [31]. Resistance is the inherited ability of a plant to survive a treatment that would kill a normal population of the same species, and the development of resistance in a plant to any form of control (chemical or non-chemical) is a process of ecological selection. The number of unique cases of herbicide-resistant weeds identified in managed turfgrass systems in the U.S. has been increasing since the late 1980s. *Poa annua* is considered the most troublesome weed in turfgrass systems and resistance is likely a factor in that ranking [13,32]. Overall, there are more cases of herbicide-resistant *Poa annua* than any other turfgrass weed [33].

A key challenge to sustaining high-quality turfgrass systems is to identify socio-economic barriers to the adoption of herbicide resistance best management practices (BMPs). This research seeks to improve understanding of these barriers by conducting a series of focus groups exploring turfgrass managers' perceptions and experience in managing herbicide resistance. The focus groups provided a systematic means of gathering perceptions and experiential knowledge from those directly engaged in managing turfgrass systems. While focus group participants' perceptions and knowledge may or may not be consistent with codified science, they likely influence weed management decisions and, therefore, are important to understand and guide broader analyses. While focus-group methods have been used to explore turfgrass and weed-management objectives and constraints [34–39]; to our knowledge, this is the first focus-group study specifically addressing herbicide resistance in turfgrass systems.

Six focus groups were conducted throughout the United States during 2019, with professionals identified by SCRI project scientists for each turfgrass sector (golf courses, sports fields, lawn care, and seed/sod production) (Table 1). The focus groups explored

whether the participants believed they had encountered herbicide-resistant *Poa annua* in their operations and if so, why they thought this might be the case; the role of non-chemical practices in weed management; challenges to effective weed management; assistance or incentives needed to implement best management practices; what sources of information participants rely on for information or advice in managing weeds; and other observations about weed management in turfgrass systems. (See Supplementary Materials for a sample focus group script).

**Table 1.** Focus group location and participants.

| Location | Date | Venue | Participant Industry | Number of Participants |
|---|---|---|---|---|
| Phoenix, AZ, USA | January 2019 | Sports Turf Managers Association (STMA) Conference | Sports Field | 5 |
| San Diego, CA, USA | February 2019 | Golf Industry Show (GIS) Conference | Golf | 7 |
| Charlotte, NC, USA | March 2019 | Turfgrass Producers International (TPI) Conference | Sod/Seed Production | 11 |
| Greenwich, CT, USA | March 2019 | Northeastern Region | Multi-sector | 6 |
| Corvallis, OR, USA | August 2019 | Oregon Seed Industry | Sod/Seed Production | 5 |
| Louisville, KY, USA | October 2019 | Landscape Professionals /Green Industry Equipment Exposition | Lawn Care | 8 |

As most of the focus groups did not include representatives of the agri-chemical industry, phone interviews with four industry representatives were also conducted in July 2019 to better understand their perspectives on the evolution of herbicide resistance. Industry representatives were selected based on recommendations from SCRI project scientists of individuals who would be able to provide thoughtful and informed perspectives on the extent of herbicide-resistant *Poa annua* and approaches to management. (See Supplementary Materials for industry interview script). Due to the diverse nature of participant industries and the different methods of information collection, no attempt was made to code or quantify responses. Results of this study have, however, been used to assist in the design of a related industry survey.

## 3. Results

The overall findings from the focus groups and industry interviews revealed that a complex set of factors influences whether managers believe they are experiencing herbicide resistance and the management options they have to intervene in the development of herbicide resistance. These dynamics vary in different regions and sectors and contribute to the "wickedness" of managing this common-pool resource. Here, we focus on several attributes of "wicked problems" that emerged from these discussions, specifically uncertainty—the differing ideas about what the "real" problem is and what is causing it—and complexity—the many interdependencies among factors and actors and the lack of a clear solution. We then turn to management options and the need for multi-faceted and inclusive strategies.

### 3.1. Uncertainty

A variety of factors contribute to uncertainty regarding whether focus-group participants and industry representatives believe that they have encountered herbicide-resistant *Poa annua*, including geographic location, management setting, effectiveness of standard weed-management practices, and the inherent attributes of *Poa annua*. A number of focus-group participants working in southern U.S. regions indicated that they had seen what they perceived to be clear evidence of herbicide-resistant *Poa annua*, while industry representa-

tives whose markets included southern areas also expressed a high level of confidence of what they perceived to be herbicide-resistant *Poa annua* developing in these regions. As one focus group participant noted, "Poa is a big problem, especially in the Southeast, more so than anywhere else. You know, we have a little bit of it in the Midwest, maybe in southern Indiana, but it is not a huge problem. Not like it is when you get down to the Carolinas and the Gulf States" (Louisville, KY, USA) (focus-group quotations are identified by their location). Another participant from this region indicated that they were "sure we have herbicide-resistant *Poa annua*" (Phoenix, AZ, USA), while another participant commented that *Poa annua* resistance was "not a problem but it is going to be pretty soon." (Phoenix, AZ, USA). In other regions, there was less clarity about whether herbicide-resistant *Poa annua* is present. Particularly in northern regions, managers generally did not believe that they had seen evidence of herbicide-resistant *Poa annua*. In northern regions, *Poa annua* is sometimes a desired species for putting greens and other uses [40], which adds to the complexity of managing this species in that region.

Another factor that influenced the focus-group participants' perceptions of possible resistance was uncertainty about whether the *Poa annua* management practices they used were effective. For example, a lack of effective *Poa annua* control could derive from problems in spraying uniformly, non-optimal timing of applications, and other factors apart from resistance. Several participants indicated that they believed they had observed evidence of resistance when they noticed clumps of green *Poa annua* after they had sprayed herbicide; one commented that "there is no way it was missed by my sprayer—it is not a strip—then there is this huge giant clump. So I know I got it [with my sprayer]" (Phoenix, AZ, USA). Another noted that "when it gets to the resistance issue, I've seen it. We know what we are doing; we know the chemistries we are using; we are making a pass with the sprayer, or hitting the Poa, right there; there is a living plant right in the middle of it. So on our end, I have no reservation that what I am doing—or was doing—was adequate to manage this certain pest for the most part and for the most part I did, but right there, it didn't. To me, that is resistance." (Phoenix, AZ, USA).

Others, however, were less certain that they had observed herbicide-resistant *Poa annua*. One commented "I can't say that I found something that was resistant. I found sloppiness in myself or maybe things that weren't in my control that made things difficult" (Phoenix, AZ, USA); another noted that "there are times I am like 'Golly, I am throwing the kitchen sink at this!' But I never said to myself I have a resistance issue. I . . . just work harder—aerification is going to be part of this, my timing going to be right, my diversification on my Ronstar (trade name for oxadiazon—a group 14, PPO inhibiting herbicide) and my PRE-M (trade name for pendimethalin—a group 3 herbicide—inhibition of microtubule assembly)." (Phoenix, AZ, USA). The Global Herbicide Resistance Action Committee (HRAC) has developed a group numbering system to identify herbicides that have the same mode of action (MOA). However, another participant noted that "maybe I have not seen what I call resistance because I think 'well, maybe it just needs a higher rate [of application]." (Phoenix, AZ, USA).

Another challenge that contributes to uncertainty around this issue is that some of the inherent characteristics of *Poa annua* make it difficult to assess whether or not herbicide resistance is developing. Participants from multiple regions commented that the adaptive nature of *Poa annua* makes it hard to determine if their management practices are adequate to control *Poa annua*, regardless of whether or not herbicide resistance might be developing. As one participant commented, "the genetic diversity of Poa has allowed it to more or less overcome every classic chemistry that has ever come out." (Louisville, KY, USA). In all regions, focus-group participants expressed widespread recognition that *Poa annua* is hard to manage due to its prolific seed production and capacity to adapt to a wide range of environments. One participant noted that *Poa annua* "has an exceptional ability to reproduce—[it] is going to introduce a tremendous genetic variety . . . this is why they recommend you rotate chemistries" (Phoenix, AZ, USA). Another participant credited the adaptive characteristics of *Poa annua* as one factor that itself could contribute to the

development of resistance, noting that, "I think it is just the sheer volume of the seed that they produce and can produce at so many different levels, that their timeframe for developing resistance compared to a lot of other weeds just—it seems like it happens fairly rapidly" (San Diego, CA, USA). Another participant commented that they "didn't observe specific resistance in the plants, it was just a matter that it is such a copious seed producer—it produces seed, multiple generations a year . . . which also brings an opportunity for it to develop resistance quickly." (San Diego, CA, USA).

### 3.2. Complexity

Herbicide rotation—rotating between herbicide modes of action (MOAs) from year to year or within a season—is among the most widespread and cost-effective techniques to reduce selection pressure and delay resistance [41]. The MOA of an herbicide is the way in which it causes physiological disruption at its target site on the plant. Yet, focus-group participants acknowledged that inadequate rotation of chemical herbicides is the main factor that would contribute to the development of resistance and generally attributed inadequate rotation to the lack of available herbicides with different MOAs for use in turfgrass systems. A complex set of factors contributes to the availability of alternative and effective herbicides, including the cost and time required to develop new chemicals, restrictions on the use of some chemicals in some sectors, and the relatively small market size of the turfgrass sector compared to production agriculture. The perceived lack of "available" herbicides for rotation may also reflect other barriers to their use, such as price, turfgrass tolerance, and regulatory or practical limits to their use in specific scenarios (e.g., golf-course putting greens or school grounds).

In the southern regions, the herbicide Specticle (trade name for indaziflam, a group 29 herbicide—cellulose biosynthesis inhibitor) was identified by several participants as the most effective control for *Poa annua* management. Several participants voiced concern about the lack of alternative chemical treatments (particularly pre-emergence herbicides, which are applied to the soil before weed seeds germinate to prevent weed seedlings from becoming established) with different MOAs that could be rotated with Specticle. One participant queried "what is the ultimate goal [of the SCRI project], to come up with a herbicide? That is what we need . . . That's the way you got to spend the money—develop a new herbicide to kill Poa . . . " (Charlotte, CT, USA). While many participants expressed hope that a new pre-emergence herbicide would be developed that could either be rotated with or replace Specticle, there was little confidence such chemistry would be developed in the near term. There was also discussion about the possible approval of methiozolin, which was being used in Korea. At the time of this focus group, this herbicide could be used on golf courses only under an experimental use permit from the EPA. It was granted full US registration for golf-course use in December 2019 (https://www.poacure.com/ (accessed on 1 October 2022)). The cost and time required to obtain a new chemical through testing and approval by regulators was widely viewed by participants as the main obstacle to development of commercially available alternatives. The challenges of developing new herbicide MOAs have been well documented [42].

Other reasons that the focus-group participants believed they did not have access to the chemicals they needed included the restriction on the use of some chemicals for certain turfgrass scenarios, such as sports fields, and the expense for companies to develop new products, given the relatively small size of the turfgrass industry compared to the overall agricultural sector. Focus-group participants and industry representatives also observed that the herbicides available for use in turfgrass management were primarily developed for use in the agronomic and horticultural crop sector, and, therefore, might not be tailored to turfgrass dynamics; as one participant put it, "everything is trickle down from ag. We get all our products from ag" (Phoenix, AZ, USA). A number of focus-group participants perceived that the relatively small scale of the turfgrass industry, compared to the agronomic crop industry, discourages turfgrass-specific research and development due to the inability of manufacturers to recover needed investment costs in discovery and commercialization,

including meeting government regulatory requirements. One participant noted that "the sod market [is] too small for chemical companies to put the money in; they're not going to do the research so it has to come from [a] university . . . because chemical companies aren't gonna do it. Because . . . we're not big enough" (Charlotte, CT, USA). Another noted, "I still think the biggest problem . . . for sod production, is that we're viewed as a tiny, tiny segment of a chemical company's sales. Those [agri-chemical company] guys are dealing with billions and billions of dollars and they just look at it and 'just a few million'?" (Charlotte, CT, USA).

One of the challenges related to the use of chemicals that are not developed specifically for turfgrass is that the label specifications are not specific to turfgrass characteristics and conditions. One participant noted that "we can't apply enough herbicide to cover us for the entire year. We max out the labels . . . [I'm] hopeful that with conventional ag beginning to look at possibly new herbicides, that some of those will trickle down to non-ag uses— hopefully to turf at some point" (San Diego). Similarly, another participant noted that "if we are going to start seeing what we are discussing—resistance—I think that is partly because we are not able to get more modes of action. So we are forced to use the same chemicals or modes of actions and start seeing resistance." (Phoenix).

Another participant shared a concern that chemicals were being taken away from use in sports fields and on golf courses because of the misuse of certain chemicals in the agricultural sector, commenting that, because the agricultural sector has "their lobbyists, they end up ok and we end up getting hammered for it." (Phoenix, AZ, USA). As another participant put it, "We feel like we make up this little section of a big piece of pie and that bigger piece [is] making some decisions that we are feeling the brunt of—because they can afford to stand in Washington DC with their big lobbies and push it down [to us]." (Phoenix, AZ, USA).

Interviews with representatives from the herbicide industry and comments from industry participants suggested that there was a high level of awareness of the challenges of managing herbicide-resistant *Poa annua* among these individuals, as well as concern about whether enough was being done by the agri-chemical industry to limit the development of resistance. Echoing [42], the costs of developing products for the turfgrass sector were also acknowledged by one industry representative, who commented that, from a company perspective, the hurdles to getting a product through EPA review and registration were getting "tougher and tougher" and that, "the bar gets set higher and higher. But then you consider the fact that it might take $300 million to develop a product from a novel compound to something that could be sold out in the industry...$300 million to invest in a new product to get it out in year 12, and you only have a 6-year, 7-year window . . . when it goes generic. And . . . [turfgrass managers] are looking at it from a cost standpoint, and I get that . . . but from a company standpoint, that is the dilemma we are in—it costs us $300 million and people want to see new chemistry come along, but they want to just go right to the generics." (Louisville, KY, USA)

The lack of cost-effective herbicides that can be rotated in treating *Poa annua* and the dim prospects that industry will develop new chemicals may be contributing to overreliance on a few available chemicals. If weeds face greater exposure to a smaller set of MOAs, this may accelerate the evolution of resistance.

A lack of understanding of the MOAs of specific chemicals may also contribute to the challenges of the effective management of *Poa annua* to avoid the emergence of herbicide resistance. Some focus-group participants commented that they believed that some managers may not be aware that certain chemicals are the same MOA and, therefore, that using them both does not constitute true herbicide rotation (San Diego, CA, USA). Use of different chemicals that, nevertheless, have the same MOA can accelerate the development of resistance to that MOA generally. As one participant put it, "I don't know that all superintendents would know that Monument (trade name for trifloxysulfuron, a group 2, ALS-inhibiting herbicide) and Revolver (trade name for foramsulfuron, a group 2, ALS-inhibiting herbicide) are the same MOA. Because a lot of times they will say 'well, I

sprayed Monument and then I sprayed Revolver and that didn't work either.'" (San Diego, CA, USA). This participant went on to say, "I was assuming we probably had prodiamine resistance, which is a Group 3. And I was like, well the property hasn't put Kerb (trade name for pronamide, a group 3, mitotic-inhibiting herbicide) down ever, so this will be great. What I didn't fully put together—Kerb is a Group 3 too! It is a very different product but it still is a Group 3 and we go out there and spray Kerb—we might as well just have thrown it in the garbage." (San Diego, CA, USA). Without a broad-based understanding of MOAs by users, the managers may be missing an opportunity when rotating to limit emergence, or may even exacerbate the emergence and spread of resistance by repeating applications of the same MOA.

Marketing practices such as Early Order Programs (EOPs) also appear to influence decision making in turfgrass management in ways that may affect the development of herbicide resistance. EOPs for pest-control chemicals reduce turfgrass managers' product costs via rebates, discounts and other monetary benefits if they order herbicides ahead of the next season of use [43]. (Manufacturers are able to offer reduced prices because these programs lower inventory costs.) In addition to a financial incentive, advance ordering via an EOP avoids potential in-season delays that could compromise a turfgrass manager's ability to treat weeds at optimal timings [44].

Industry representatives interviewed regarded EOPs as having a significant influence on decisions about which chemical herbicides are used in turfgrass weed management. Focus-group participants from the lawn-care sector (the only focus group where a question about EOPs was posed) agreed that the program increasingly influenced decision making regarding product choice. There was some concern that EOPs might influence managers to choose a weed-control product even when it is not the best choice. For example, if particular chemistries are already paid for, they are "sunk costs" for managers and managers may then apply what they have on hand without due consideration of how those compounds may affect resistance.

In the case of sports fields, another factor that focus group participants identified as a challenge to managing turfgrass effectively is gaining access to fields to implement management practices at the appropriate time, as scheduling of games and practice sessions are often given a higher priority by those with authority to make such decisions. As one participant noted, "The bottom line is we have to react to the event schedule and we don't have the ability—even when [a manager] knows and he had done his due diligence—'I have my reentry [interval as specified by label] and this is what I am going to do'. Well guess what, when [a coach or other decision maker] decides what he is going to do [the turfgrass manager] has no ability to stop that and all of a sudden it is his fault." (Phoenix, AZ, USA).

The focus groups also explored the role of "non-chemical practices" in managing turfgrass, as the integration of approaches that do not rely on the use of herbicides may reduce the "selection pressure" that drives resistance. (The term "cultural practices" is also sometimes used, though these two terms may not always be synonymous). "Non-chemical practices" is used here instead of "cultural practices" as the former appeared to better describe the content in focus-group discussions. We note there is some ambiguity in what falls under each category. Focus-group participants noted that non-chemical practices were an important complement to herbicide applications, but that these practices could not substitute completely for the use of herbicides. As one participant noted, "Chemistry is important—we need that [and] rotating . . . is important to use this tool correctly. In our industry, focusing on mowing is the most important thing you do, aerifying is the second most important thing you can do. That you are fertilizing appropriately and that you understand that the biology in your soil is important." (Phoenix, AZ, USA).

Another participant commented that, "Probably the most effective thing we have against Poa resistance is fraise mowing" (Phoenix, AZ, USA), while a sod producer commented that *Poa annua* "just never gets established because of the tillage, the constant tillage . . . we are constantly tilling the ground. I think that is how we address it the easiest way."

(Charlotte, NC, USA). In other situations, labor-intensive practices such as hand-weeding are used; one participant noted that, "[as] a last resort they're hand-pulling it—they are plucking it out the greens." (Charlotte, NC, USA). Another sod producer agreed: "Yeah, we hand pull it in the winter . . . when we have labor to do it. Fields you might want to harvest. It takes forever, but we pull it." (Charlotte, NC, USA).

Strengthening other desirable grasses to compete better with *Poa annua* was another strategy that some managers used; one participant commented that they selected for "the most aggressive, most competitive types [of grasses] to try to constrict [*Poa annua*] out." (Charlotte, NC, USA). Another participant noted that they were "experimenting with some clovers and some other things too to see if we can get some growth and anything that can be left in the soil, both from a soil structure standpoint and also if it's got nitrogen in there, all the better . . . just to try to get that vertical growth for the [Kentucky] bluegrass so at least we can compete with the Poa more quickly in the season."(Charlotte, NC, USA). Crop rotation was also used to control *Poa annua* in select turfgrass production scenarios; for example, participants in one focus group discussed using blueberries or radishes in rotation with grass seed production (Corvallis, OR, USA).

As noted above, the effective implementation of both non-chemical practices and herbicide applications can be limited by challenges such as access to fields at the appropriate times as well as by cost of labor to implement practices such as hand weeding. In addition, there was some discussion of preventative approaches that could help reduce the spread of *Poa annua*—whether herbicide-resistant or not—across properties, such as cleaning golf shoes before entering a course and cleaning equipment between properties or sites.

The term "multiple resistance" refers to the resistance of a plant to multiple and different herbicide MOAs. Increased instances of multiple resistance for weeds in agronomic and horticultural crops narrows the range of effective herbicides and may increase risks of yield loss and increased production costs [45]. Some focus-group participants indicated they had experience with *Poa annua* exhibiting apparent resistance to multiple MOAs; while these participants did not yet consider this to be a critical problem, some voiced concern that it might become a greater challenge over time.

If multiple resistance is being observed, this may motivate turfgrass managers to explore additional strategies to manage resistance, including cooperative approaches; one recent study of farmers' views toward managing herbicide resistance found that perceptions of multiple resistance increased their concern about resistant weed migration and their receptivity toward cooperative management approaches [46]. This last point suggests options for management interventions. Cooperative management approaches including producer groups, university specialists, and the private sector have been implemented specifically to address herbicide-resistant weeds in agronomic crops [47–50]. Other examples of cooperative weed-management approaches include Areawide Invasive Weed Programs and Cooperative Weed Management Areas (see [10] for discussion of the structure and function of these types of organizations).

## 4. Discussion: Management of *Poa annua*: Acknowledging Uncertainty and Embracing Complexity

The focus-group explorations revealed that the development of herbicide resistance in *Poa annua* is indeed a wicked problem, whose causes are "convoluted by myriad biological and technological factors, and are fundamentally driven by the vagaries of human decision-making" [5] (p. 552). The focus-group discussions also indicate that herbicide resistance in *Poa annua* is not yet viewed as a crisis, suggesting that there may be time to formulate a strategic plan and related interventions to manage the development of resistance. Addressing the common-pool nature of resistance adds another layer of complexity as the decisions involve multiple sources of uncertainty, including strategic human interactions. The complex factors influencing herbicide resistance in *Poa annua* point to important management considerations, including the need for multiple strategies and solutions reflective of specific contexts, the importance of developing integrated solutions that recognize the

interconnectedness of socio-ecological systems, and the importance of these solutions being adaptive to different and evolving circumstances.

Developing an integrated and adaptive approach that recognizes the multi-faceted nature of the challenge will be critical given the complexity of the issue and the dynamics we have described. Strategies to address this challenge may be informed by work performed on both approaches to "wicked problems" as well as insights into the management of complex natural-resource management systems and common-pool resources. For example, Allen's exploration of chemical policy as a "wicked problem" [2] suggests that strategies that are collaborative, adaptive, and that take a systems approach may be more effective in addressing these challenges. Allen [2] notes that "While the complex and evolving nature of wicked problems means that they rarely can be solved . . . policies that are more collaborative - especially when there are multiple stakeholders and the power among stakeholders dispersed—tend to be more effective in engaging the range of actors who need to be involved in addressing a particular issue [51] (p. 102). The variety of actors who have a stake in that effective management of *Poa annua* suggest that it may be important to explore collaborative approaches to addressing this issue. Allen [2] goes on to note that because "wicked problems are by their nature imperfectly understood, policies that are innovative, flexible and adaptable—that acknowledge the incompleteness of current knowledge and the uncertainty inherent in understanding of complex issues—may be better able to evolve as new information becomes available." [2] (p. 102). This observation suggests that it will be important to invest in ongoing learning and feedback loops to assess how management strategies need to be adapted over time. Finally, Allen notes that understanding and developing solutions to wicked problems requires "holistic systems thinking in order to adequately grasp the interrelationships between the multiple causal factors and policy objectives such problems encompass [4]." [2] (p. 102). While taking a systems approach that engages a broad spectrum of stakeholders and that remains open to new information and knowledge over time can be more challenging than investing in targeted technical fixes, over the long term such an approach is likely to be more effective in managing this kind of complex challenge.

Allen et al. [52] suggest a number of factors that are important to take into account when trying to address common-pool resources. Of these factors, those that seem particularly relevant here are the importance of "people and place", embracing complexity, actions that are integrated across scales, and designing for resilience. We use these principles to explore possible management strategies to address the development of herbicide resistance in *Poa annua*.

### 4.1. Understanding People and Place

The focus groups revealed that the dynamics around herbicide resistance in turfgrass vary by region and sector, and that a variety of factors shape managers' decisions. The factors that may contribute to the development of resistance and that confound efforts to determine whether treatments are effective can make it difficult to identify what kinds of management approaches may be needed in a specific case. The variations across different zones in terms of both *Poa annua's* viability and the available means of treatment mean that management options also vary widely across the country. Jussaume and Ervin [7] advise that "managing resistance must be viewed as a wicked problem with no standard template across regions." [7] (p. 2). How good or bad a solution is for addressing a wicked problem varies depending on "local context, economic parameters, social costs, etc." [7] (p. 3). Elinor Ostrom similarly notes with regard to common-pool resources that "If the initial set of rules established by the users, or by a government, are not congruent with local conditions, long-term sustainability may not be achieved" [53] (p. 421).

As the dynamics of managing *Poa annua* vary by region and turfgrass sector, taking different contexts into account in developing management strategies will be an important consideration. Several approaches identified by focus-group participants may be worth considering. For example, some participants thought it would be helpful to have

more specific multi-year programs that are both sector- and species-specific; ideally, such multi-year "programs" would provide guidance on management practices to avoid the development of resistance. As one participant put it, "Maybe it would be useful from an end user standpoint if some of the scientists could begin to drill multi-year, pre- and post- emergent programs, that could help not only with weed control but also minimize resistance of some of the weeds we are targeting. There would have to be some caveats into all these—[turfgrass] type, specific weeds, and it is not just herbicides, cultural and fertility aspects to this as well . . . A potential deliverable of this would be regionalized resistance management, pre- and post-emergent herbicide programs to help minimize the potential for long term [resistance challenges]—because who knows what is going to be the next weed that we are going to be worried about?" (San Diego, CA, USA).

### 4.2. Understanding and Embracing Complexity

As wicked problems have no single or easy-to-identify cause, there is a high level of uncertainty when it comes to understanding the problem and proposing solutions [7]. As Jussaume and Ervin note, not only is it "difficult to identify a single cause, but the selection of any single cause can be arbitrary and may limit our ability to recognize that there were multiple causes, some of which may be unknown, and multiple potential solutions, all of which will need to be considered over time." [7] (p. 5). Elinor Ostrom has similarly noted that a "core challenge in diagnosing why some [socio-ecological systems] are sustainable whereas others collapse is the identification and analysis of relationships among multiple levels of these complex systems at different spatial and temporal scales? . . . we must learn how to dissect and harness complexity, rather than eliminate it from such systems" [53] (p. 420).

The focus-group participants and agri-chemical industry representatives identified a complex set of factors that contribute to the development of herbicide resistance in turfgrass systems, including inadequate rotation of MOAs, lack of access to effective alternative chemicals, as well as a possible lack of understanding regarding MOAs and what different products actually offer in terms of rotation. Challenges related to integrating non-chemical practices may also affect the development of resistance, including constraints on gaining access to fields at the appropriate times for such interventions, cost constraints related to practices such as hand weeding, and weather conditions that may preclude certain kinds of practices at the appropriate time. These factors, combined with the challenge of managing *Poa annua* due to its prolific seed production and related adaptability to different environments, may confound attempts to determine whether herbicides are in fact effective as well as whether resistance is developing.

Ignoring one or more important contributing factors in favor of only one possible cause may truncate the exploration of how the whole system works to produce or control resistance and, therefore, may overlook important opportunities to intervene. If resistance is due to multiple factors, e.g., lack of R&D on new chemical treatments or non-chemical practices, management time constraints at crucial weed growth periods, etc., then any single focused intervention may have little overall impact.

The multiple potential causes of resistance in particular settings suggests using active adaptive management strategies to learn about ways to reduce uncertainty; such strategies would vary across regions, given the different dynamics in different locations. Active adaptive management has the explicit goal of learning about system dynamics rather than just observing changes in the resource conditions, i.e., resistance patterns, as under passive adaptive management [43]. Trials can incorporate the effects of various management practices to learn more about the socio-ecological systems that govern the evolution of resistance in specific situations. For example, turf managers, in collaboration with scientists from public and private organizations, could conduct experiments in their operations using combinations of chemical practices (e.g., herbicides and growth regulators), and non-chemical practices (e.g., mowing height and irrigation) to evaluate their costs and ability to control the spread of resistant weeds such as *Poa annua*.

### 4.3. Action and Integration at All Scales

Related to the complexity noted above, because wicked problems often interface with each other, there is a "need to develop holistic management systems that recognize how subcomponents of that system are interconnected." [7] (p. 5). As Shaw [5] notes, "Herbicide resistance will only be managed through the combined efforts of all parties (e.g., industry, university, government, retailers/dealers, and consultants). Everyone involved must understand the role each individual can play and work collectively to achieve the outcomes we all desire if we are to overcome this wicked problem. All involved must also hold each other accountable for actions. We must not only think differently; rather, we must act differently." [5] (p. 557).

One of the most hopeful findings from the focus groups is that herbicide resistance is not yet viewed as a crisis. This suggests that effective proactive control schemes could avert significant future damage caused by resistance spreading in turfgrass systems. This will take a concentrated effort amongst motivated turfgrass managers, product suppliers, weed scientists, government and industry associations, professional societies, and the media to champion resistance management throughout their communities [15,54].

### 4.4. Design for Resilience

As the evolution of resistance in weeds is continual, there are no final stopping points or permanent solutions in resistance management. Consequently, weed managers will need to continually adapt to changing circumstances and to develop new strategies. While new MOAs can aid in adaptation to herbicide-resistant weed species, evolution means that they will only be temporary solutions [7]. This lack of a clear end point is a classic feature of wicked problems. One response to wicked problems is pressure to find a quick and simple solution. Yet, strategies will in fact need to be ongoing and adaptive [7]. These observations draw attention to the importance of ongoing learning and feedback loops to help ensure that strategies are adapted and adjusted as conditions and knowledge evolve.

One focus-group participant noted, "[herbicide resistance] is a long term issue that needs to be managed but unfortunately it is a long term problem that I don't think is hurting anyone bad enough yet—and that is the problem, is that our society today changes so quickly because of the technology and apps . . . everybody is just thinking acutely, they are not thinking about years down the road and even if it is brought up and proven . . . that is someone else's problem. And that goes back [to] culture creation—whatever happened to 'this is our community, it matters, I am going to be a steward, whether I am going to move to another state, but that doesn't mean I can abuse where I am because I am not going to be here anymore?' And I think that has been lost a bit." (Charlotte, NC, USA).

Another participant noted that they should try "to take the options that are out there and utilize those as much as we can in rotations, not just within the year but within multiple years, so we try to minimize the potential for resistant weeds in the future." (San Diego, CA, USA).

The availability of and access to information and to developing knowledge about approaches and options is an important element in designing for resilience. For this reason, it may be helpful to identify the following:

- Who (or what stakeholder groups have?) has information on proactive management options?
- What information sources are best for effective communication to different sectors and end users?
- Who can provide guidance on long-term strategies?
- How can this information best be disseminated?

The main sources of information that focus-group participants relied upon for guidance on management practices include universities (particularly cooperative extension programs), chemical distributors, herbicide manufacturers' representatives, and the internet, as well as colleagues and regional or national sector associations (e.g., Sports Turf Management Association (STMA) or Golf Course Superintendents Association of America

(GCSAA)). These information sources are similar to those used by producers of agronomic crops. One study found that producers receiving information from cooperative extension and universities were more concerned about herbicide resistance than those who relied on chemical dealers and farm input retailers [55]. A follow-up survey a few years later obtained similar results [56].

Several participants also commented that they conduct their own "due diligence" by gathering information from multiple sources to assess what products or practices might be effective before adopting a particular strategy. As one participant noted, "I do my due diligence—I will call multiple sales guys and say 'This is what I am seeing, what are my options?' Obviously they will endorse their product—then I do my due diligence—go to manufacturer's website—investigate products, rates, what are they controlling. I am going to the manufacturer's website—and then we have the advantage of having Extension people in College Station and I will pick their brains. But even then I do the same things. I go back to the manufacturer and do my own due diligence." (Phoenix, AZ, USA).

While turfgrass managers may seek out universities and cooperative extension programs for information, not all turfgrass managers may have access to the same level of support from university systems as those who participated in these focus groups. Unequal funding across a range of locations and within sectors means that many turfgrass managers are not members of the associations described here, or they may not be able to attend conferences and expos where the most up-to-date information is shared. Likewise, proximity to university and cooperative extension specialists varies greatly across regions and states. For instance, not every state has extension turfgrass weed scientists, nor are county and regional agents able to assist with the complex and disparate nature of the multi-sector turfgrass industry.

Manufacturer representatives and chemical distributors were also identified as a source of information, though there were some concerns about whether manufacturer representatives were biased toward recommending their own products. Distributors who carried a range of products were considered to be more objective about the relative effectiveness of different products; however, it is possible that distributors in some cases promote products that offer greater immediate financial return rather than effective weed control.

Multiple participants expressed interest in obtaining access to information about documented events of resistance and guidance on best management responses to such events. A number of participants indicated that they felt there were opportunities to provide additional information about resistance through national or regional sector associations, either at conferences, via webinars, or through other means of communication. This sentiment was echoed in a subsequent study by a focus group specifically targeting the lawn-care industry [54].

However, while additional national or regional efforts may be necessary, these alone may be insufficient to alter turfgrass managers' behavior. Asmus and Schroeder [57] stress the need for greater collaboration with land managers (in their case farmers), consultants, and dealers, to craft messages that are truly useful. One person succinctly identified the following key actions that would be helpful in managing the development of herbicide resistance:

- Information about proactive treatment options to address the resistance "that we are seeing now".
- Communication that "we do have an issue" and what people need to be looking for.
- Long term strategies on how to manage "so we don't have issues in the future".

### 4.5. Information Is Important but Problems Are Solved by People

Developing comprehensive and collaborative strategies that cross ownerships and sectors can be challenging as there is no one umbrella organization or network that encompasses the turfgrass management sector. Some social studies have identified needs in pesticide training and continuing education for groundskeepers. For instance, work by [58]

suggests that 36% of the school districts in the USA employ individuals that perform pesticide applications without any training, licensing, or certification. A later study determined that grounds employees wanted information on weed management and herbicide use more than any other training topic [38]. Stock et al. [59] found that the annual turnover rate for a grounds employee was 24.3%, which poses a considerable challenge for the development of institutional knowledge concerning resistance identification and management. Furthermore, only 65.4% of employees utilized extension herbicide recommendations provided by their local university, and an even smaller fraction of new employees (19.5%) used these recommendations. This would suggest that new, relatively inexperienced employees are regularly making weed-management decisions without the available extension material. A recent analysis of public land managers in Oregon, Indiana and New Jersey found that reducing herbicide use was a potential advantage produced by transitioning to low input turfgrass [39]. While these previous studies have documented a need and interest in pesticide training and low input turfgrass systems, to date no national social study of weed management or herbicide resistance in turfgrass systems has been conducted.

## 5. Conclusions

The development of herbicide-resistant *Poa annua* in turfgrass systems is complex and evolving. The focus-group participants suggested multiple factors that may contribute to the development of resistance in different turfgrass sectors, including challenges related to having access to turfgrass fields (particularly in sports) at the appropriate times to apply treatments, possible lack of understanding regarding different MOAs and proper rotation procedures, costs associated with some non-chemical practices, and access to alternative products that can be rotated to help mitigate herbicide resistance.

As noted previously, while focus groups have been usefully employed to examine resistance management for agronomic crops, this has not been carried out for turfgrass systems; these focus groups address that gap. However, while such qualitative methods can shed light on important aspects of herbicide-resistance management in turfgrass systems, a more quantitative assessment would be a worthwhile complement to these focus-group results. In fact, these focus-group findings have been used to help develop a survey for just such a quantitative analysis and have informed the formulation of hypotheses to be tested with the broader survey data. Such hypotheses relate to, for example, extension-program participation effects, level of understanding of different MOA's, how awareness and concerns about *P. annua* affects weed management, and opportunities to build a more holistic systemic understanding of resistance management in turf systems. The findings from this survey will help broaden understanding of the opportunities and challenges related to herbicide-resistance management across the USA.

Given the complex and evolving nature of this challenge, developing strategies that take into account the dynamics of different regions and sectors; offer opportunities to integrate new knowledge and learning; and engage managers, extension programs, industry partners and others in a collaborative manner offers the best opportunities to manage and mitigate the development of herbicide resistance in turfgrass systems.

**Supplementary Materials:** A sample focus group script and industry interview script are included in Supplementary Materials. The following supporting information can be downloaded at: https://www.mdpi.com/article/10.3390/su142013399/s1.

**Author Contributions:** Conceptualization, J.H.A., D.E.E. and G.B.F.; Data curation, J.H.A.; Formal analysis, J.H.A., D.E.E. and G.B.F.; Funding acquisition, M.V.B.; Investigation, J.H.A.; Methodology, J.H.A., D.E.E., G.B.F., J.T.B., J.D.M., R.G.B., A.J.P., M.T.E., J.E.K. and A.R.K.; Project administration, D.E.E. and M.V.B.; Resources, J.H.A., D.E.E., G.B.F., J.T.B., J.D.M., R.G.B., A.J.P., M.T.E., T.W.G., L.B.M., P.E.M., J.E.K., S.D.A., A.R.K., J.B.U., J.S.M. and M.V.B.; Supervision, D.E.E.; Validation, J.H.A.; Writing—original draft, J.H.A., D.E.E. and G.B.F.; Writing—review and editing, J.H.A., D.E.E., G.B.F., J.T.B., J.D.M., R.G.B., A.J.P., M.T.E. and T.W.G. All authors have read and agreed to the published version of the manuscript.

**Funding:** This project was funded by the USDA-NIFA Specialty Crops Research Initiative (SCRI) program (award #: 2018-51181-28436).

**Institutional Review Board Statement:** The study was conducted according to the guidelines of the Declaration of Helsinki and approved by the Institutional Review Board of Portland State University (protocol code 184844, approved on 3 December 2018, 29 May 2019, and 8 August 2019).

**Informed Consent Statement:** Informed consent was obtained from all subjects involved in the study.

**Data Availability Statement:** The data presented in this study are available on request from the corresponding author. The data are not publicly available due to human subject review requirements and protections.

**Conflicts of Interest:** The authors declare no conflict of interest.

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
