# Peer review of "Herbicide-Resistance in Turf Systems: Insights and Options for Managing Complexity"

_sustainability, doi:10.3390/su142013399_

Round 1

Reviewer 1 Report

Regarding the services provided by turfgrass systems are very important the carbon sequestration, oxygen production and air purification by capturing the particles in suspension. (see first paragraph from background and methods, lines 122 - 129).

In the following, is mentioned that herbicides-resistant weeds ”can degrade turfgrass quality and diminish these environmental, economic, and social benefits over time” (lines 131-132). The observation regarding this phrase is that weeds are diminishing turfgrass quality and economic benefits, maybe in a less measure the social benefits are affected related with the previous one. But, the environmental quality benefits of turfgrass arent affected by weeds, because they have the same environmental function as grasses. They aren't desired in turf mainly from aesthetical purposes and due to the quality standards of turfgrass as product.

The interview was based on a questionary? The focus groups answers can be quantified in a way for an interpretation more than empirical?

In line 212 is a point improperly positioned and the quotation marks are missing in the sentence "Another noted that .when it gets to the resistance issue, I’ve seen it.", thus at the end of the citation.

Lines 364-365 have a bibliographical reference that shall be formatted according with the instructions for authors.

From line 462 to 473 is a very long quote that shall be integrated in the discussion more properly in relationship with the results.

In some lines the latin name Poa annua isn't in italic format.

See line 512 "..., “(n)ot..."

I suggest that long quotes the from text to be integrated effectively from the perspective of the present article, not as copy paste.

The discussion section has some long copy-paste quotes, even from the same source, only the page differs. This section shall be seriously revised regarding the cited literature integration.

Author Response

Please see attachment for responses to all reviewers' comments

Reviewer 2 Report

The present article “Herbicide-Resistance in Turf Systems: Insights and Options for Managing Complexity  deals an interesting  theme of  management of Herbicide resistant (HR) Poa annua .The article is well written and cover broad  aspect.

Author Response

Please see attached for response to all reviewers' comments.

Reviewer 3 Report

The manuscript entitled “Herbicide-resistance in Turf Systems: Insights and options for managing complexity” by Allen et al discussed herbicide resistance (HR), which poses a risk to sustainably managing U.S. turfgrass systems. The authors described six focus groups conducted throughout the United States, to understand socio-economic barriers of HR management practices. 

The minor concerns the reviewer have are shown below:

1 Line 60: The Table 1 should be appropriately formatted, according to the guideline from the journal

2 Line 79, Line 86, Line 90/93/96:  for the chapter/book citation, it is not necessary to list the page number in the main texts

3 Line 170: the reviewer can not find appendix B in the manuscript

4 The quality of clarity for the texts can be improved. The author should proof-read the manuscript. 

Author Response

(The authors gave the same response as above.)
